# Identification of Polyphenols from Coniferous Shoots as Natural Antioxidants and Antimicrobial Compounds

**DOI:** 10.3390/molecules25153527

**Published:** 2020-08-01

**Authors:** Marcin Dziedzinski, Joanna Kobus-Cisowska, Daria Szymanowska, Kinga Stuper-Szablewska, Marlena Baranowska

**Affiliations:** 1Department of Gastronomy Science and Functional Foods, Faculty of Food Science and Nutrition, Poznan University of Life Sciences, 60-637 Poznan, Poland; joanna.kobus-cisowska@up.poznan.pl; 2Department of Pharmacognosy, Poznan University of Medical Sciences, Swiecickiego, 60-781 Poznan, Poland; daria.szymanowska@up.poznan.pl; 3Department of Chemistry, Faculty of Wood Technology, Poznan University of Life Sciences, 60-625 Poznan, Poland; kinga.stuper@up.poznan.pl; 4Department of Silviculture, Faculty of Forestry, University of Life Sciences, 60-637 Poznan, Poland; marlena.baranowska@up.poznan.pl

**Keywords:** bioactive compounds, phytochemicals, antioxidant and antimicrobial properties, polyphenols, coniferous trees, shoots

## Abstract

Currently, coniferous shoots are almost absent as a food ingredient despite their wide availability in many parts of the world. The aim of the study was to assess and compare the composition of selected plant metabolites, evaluate the antioxidant and antimicrobial properties of selected shoots collected in 2019 from the arboretum in Zielonka (Poland), including individual samples from *Picea abies* L. (PA), *Larix decidua* Mill (LD), *Pinus sylvestris* L. (PS), *Pseudotsuga menziesii* (PM) and *Juniperus communis* L. (JC). The present work has shown that aqueous extracts obtained from tested shoots are a rich source of phenols such as caffeic acid, ferulic acid, chlorogenic acid, 4-hydroxybenzoic acid and many others. Obtained extracts exhibit antioxidant and antimicrobial properties in vitro. The highest sum of the studied phenolic compounds was found in the PA sample (13,947.80 µg/g dw), while the lowest in PS (6123.57 µg/g dw). The samples were particularly rich in ferulic acid, chlorogenic acid and 4-hydroxybenzoic acid. The highest values regarding the Folin-Ciocâlteu reagent (FCR) and ferric reducing ability of plasma (FRAP) reducing ability tests, as well as the total flavonoid content assay, were obtained for the LD sample, although the LD (14.83 mg GAE/g dw) and PM (14.53 mg GAE/g dw) samples did not differ statistically in the FCR assay. With respect to free radical quenching measurements (DPPH), the PA (404.18-μM Trolox/g dw) and JC (384.30-μM Trolox/g dw) samples had the highest radical quenching ability and did not differ statistically. Generally, extracts obtained from PA and PS showed the highest antimicrobial activity against tested Gram-negative bacteria, Gram-positive bacteria and fungi.

## 1. Introduction

The demand for plant health products is increasing worldwide [1]. Currently, coniferous shoots are almost absent as a food ingredient despite their wide availability in many parts of the world. The exceptions are the common juniper, whose berry-like cones are a spice valued in Europe and pine shoots [2]. Products with pine shoots are available on the market, including pine shoot syrup, beer made with pine shoots or herbal teas. Despite this, products with shoots are currently not very popular [3,4]. However, these raw materials were often used in folk medicine in the past, among others in ancient Rome or traditional Chinese and Islamic medicine. Bark, shoots and resins were used as a panacea for, e.g., diseases of the urinary tract, digestive tract, nervous system, respiratory and skin diseases [5,6,7]. Research carried out in recent years has confirmed that compounds present in conifer shoots exhibit therapeutic effects, and shoots are rich sources of polyphenols and have antioxidant properties [8,9]. Coniferous shoots are a particularly rich source of terpenoid hydrocarbon, pinene, in the form of alpha and beta isomers, which as one of the main components of the resin belongs to water-insoluble fraction. Alpha- and beta-pinene may serve as precursors of aromatic compounds in food production, they are also components of renal and hepatic drugs [9]. In rodent studies, alpha-pinene showed, inter alia, gastroprotective, analgesic and anti-convulsive properties; it has also exerted therapeutic effects in some cancers and allergies [10,11,12,13]. Compounds present in conifers also show antioxidant and reducing effects [14]. New studies have indicated that functional foods with antioxidant potential can be crucial in maintaining health. Excessive increase in free radical concentration due to endogenous and exogenous factors in the body may be conducive to damage to, i.e., biologic structures such as DNA, lipid membranes and proteins [15]. Cell oxidation–reduction homeostasis is one of the most important elements regulating the body’s functions at the molecular level, its disorders can promote dysfunctions, especially in terms of enzymatic activity [16].

Currently, the state of knowledge about the properties and applications of coniferous components is incomplete, thus far no experiments have been conducted on the use of the most popular coniferous trees in food, and these raw materials are a promising component not only of drugs or dietary supplements, but also functional food, which is currently one of the fastest-growing food market segments [17].

The aim of the study was to assess and compare the composition of selected plant metabolites, evaluate the antioxidant and antimicrobial properties of selected shoots samples from various conifers, including *Picea abies* L. (PA), *Larix decidua* Mill (LD), *Pinus sylvestris* L. (PS), *Pseudotsuga menziesii* (PM) and *Juniperus communis* L. (JC).

## 2. Results

### 2.1. Shoot Commodity Assessment

The tested shoots samples differed visually significantly (Figure 1). As shown in the Table 1, needles obtained from the shoots of individual trees were of different shapes and dimensions, the longest needles had the PS sample (54.84), while the shortest the JC sample (8.74). Shoots differed in color measured in the CIELab space. The PM sample had the highest brightness (L* = 33.74) while the PA sample was the darkest (L* = 26.09). In terms of parameter L*, the JC, LD and PS samples did not differ statistically. The values a* and b* expressed the color in the range from green to red and from blue to yellow, respectively; the highest value for the parameter a* was found in the PA sample (8.26), the lowest in JC (−5.25). The highest b* value was found in the JC sample (24.48), the lowest for PS (10.73). Samples differed significantly in terms of dry weight except for the PM and PA samples. The top-level dry weight content, significantly higher than in the other samples, was recorded for the PS sample. Extracts from dried shoots did not differ significantly in osmolality. The PA sample freezing point was the lowest (0.09), but the differences between the samples were small.

### 2.2. Phytochemical Shoot Content

There was a wide variation for other phenolic acids tested (Table 2). The highest sum of the studied phenolic compounds was found in the PA sample (13,947.8 µg/g dw), while the lowest in PS (6123.57 µg/g dw). The samples were particularly rich in caffeic acid, ferulic acid, chlorogenic acid and 4-hydroxybenzoic acid. The highest variation was found in the case of ferulic acid, where in the PM sample it was 5002.20 µg/g dw and 1129.85 µg/g dw for PA. Salicylic acid, naringenin, vitexin, rutin, quercetin, apigenin, kaempferol, and luteolin were present in very low concentrations in all samples.

### 2.3. Antioxidant and Antiradical Extract Properties

Antioxidant and antiradical properties of extracts obtained from coniferous shoots were examined using spectrophotometric methods (Table 3). The highest values regarding the FCR and FRAP reducing ability tests, as well as the total flavonoid content assay, were obtained for the LD sample, although the LD (14.83 mg GAE/g dw) and PM (14.53 mg GAE/g dw) samples did not differ statistically in the FCR assay. With respect to free radical quenching measurements (DPPH), the PA (404.18-μM Trolox/g dw) and JC (384.30-μM Trolox/g dw) samples had the highest radical quenching ability and did not differ statistically.

### 2.4. Antimicrobial Screening

The effect of water extracts from against indicator microorganisms of both Gram-positive and Gram-negative bacteria as well as mold and yeast was studied. The results obtained are summarized in Table 4. The highest antimicrobial activity was shown for PA extract against *P. aeruginosa* (32 mm), while the lowest activity was shown for PM extract against *S. aureus* (2 mm) and *L. fermentum* (3 mm). Generally, extracts obtained from PA and PS showed the highest antimicrobial activity, against Gram-negative bacteria, Gram-positive bacteria and against fungi, but the growth inhibition zone is significantly smaller in case of fungi than in tested bacteria.

Conducted correlation analysis of antioxidant assays, antimicrobial parameters and phenolic compound showed a small number of statistically significant correlations between different assays and compounds (Figure 2). The strongest statistically significant negative correlations are observed between FCR assay and inhibition of *Aspergillus* sp., between total flavonoid content assay and chlorogenic and sinapic acid content. Strongest statistically significant positive correlations are observed between DPPH assay and caffeic acid content.

## 3. Discussion

The use of raw materials such as coniferous shoots can contribute to the development of a functional food sector. Food enriched with ingredients containing active phytochemicals allows reducing the incidence of diseases, which is desirable [18,19]. Thus, the production of inexpensive, natural health-promoting products is of great importance in the current socio-economic situation of many countries in the world, especially in the context of the incidence of civilization diseases [20]. Many products perceived as health-promoting are known in natural medicine, including coniferous shoots, which are not currently widely used in this field. In this publication, individual samples of shoots of selected popular conifers, i.e., *Picea abies* L.; *Larix decidua* Mill; *Pinus sylvestris* L.; *Pseudotsuga menziesii*; *Juniperus communis* L. were characterized. It was shown that the content of polyphenolic compounds depended on the species, which significantly contributed to the antioxidant capacity.

Visual perception is extremely important for consumers when selecting food products in terms of attractiveness and perception of health-promoting properties The use of plant raw materials as ingredients in food products can affect not only the taste and aroma but also their color, thus is important to assess raw materials in this respect [21]. In the present study, it was noticed that the PA sample, which was the darkest, i.e., with the lowest parameter L*, was characterized by, among others, the highest total content of phenolic compounds measured by HPLC and the strongest ability to quench free DPPH radicals. A relationship between phenolic compounds and color hue was observed, i.e., in studies on honey, where darker samples were richer in these compounds [22]. However, it should be noted that some compounds, including anthocyanins and lycopene, can give similar color to raw materials, but they differ significantly in antioxidant capacity. The relationship between color and antioxidant content should not be generalized to all raw materials, but it can be stated that differences in color usually mean different content of bioactive compounds [23].

The tested shoots, in addition to the color, differed noticeably in length, shape and dry matter content, which had an impact on the content of bioactive compounds in the extracts. In the current study, all raw materials were dried under the same conditions, although earlier studies indicated that the drying process significantly affected the content of bioactive compounds as well as antioxidant and antimicrobial properties of coniferous shoots [8]. In the present study, distilled water was used for extraction, which allowed to obtain a high content of phenolic compounds in extracts that had high antioxidant potential, although water is not the most effective solvent. As many authors have shown earlier for vegetable raw materials, such as *Pistacia terebinthus* L. or *Limnophila aromatica*, methanol and acetone are much more effective solvents, allowing obtaining extracts with higher antioxidant potential and a higher concentration of bioactive compounds [24]. However, evaluation of the efficiency of water extraction allows us to determine, from the point of view of practical application, whether raw materials could be components of functional products in the future, i.e., infusions, beverages, smoothies or yogurts [25].

The extracts obtained differed significantly in the content of phenolic compounds, however, 4-hydroxybenzoic acid, caffeic acid, ferulic acid and chlorogenic acid were dominant among most the trees studied. Conducted statistical analysis showed strong positive correlation between total content of phenolic compounds and sinapic and chlorogenic acid, but strong negative correlations between these compounds and total flavonoid content assay. This trend may be caused because the amounts of phenolic acids are higher than flavonoids and used methods in present study may not have covered the measurement of all individual phenolic compounds [26] In study of cones and berries from 16 different Turkish coniferous species authors used 95% acetone as an extractant and also identified many phenolic compounds, such as 4-hydroxybenzoic acid, 3,4-dihydroxybenzoic acid, catechin, the total amount of phenolics was determined in range of 60–6390 µg/g dw depending on the species, which is lower than in present study [27]. On the other hand, in study conducted on needles from Norway spruce, authors used 95% ethanol for extraction and detected the same compounds as in present study, i.e., chlorogenic acid, gallic acid, kaempferol, quercetin but in concentrations below the quantification limit of about 0.02 µmol/g dw. The dominant compound, similarly to the previously mentioned study was catechin, which was not tested for in this study [28], Phenolic compounds play protective functions in plants, but they can have health-promoting effects in the human body as well as positively affect the quality and safety of food products. The study of Raitanen et al. demonstrated that tannins from pine and spruce bark protected meat snacks against fat oxidation without affecting smell and taste [29]. Many studies have demonstrated the antioxidant activity of phenolic compounds, recent findings also point to possible genoprotective and neuroprotective effects, as well as those reducing the risk of metabolic diseases, cardiovascular diseases and cancers [30]. All of the tested extracts also showed antioxidant and reducing properties demonstrated by DPPH, FRAP and FCR methods. The highest DPPH inhibition was demonstrated for the PA and JC samples, they also had the highest total phenolic content. Studies by other authors have confirmed the strong antioxidant activity of extracts obtained from juniper and pine, although there is no such data on other conifers [31]. Differences between the samples were significant in the FRAP and FCR tests. It is concluded that differences between individual methods may be due to different reactivity of individual compounds in the extracts, including compounds that have not been tested in this publication. FCR and FRAP are non-specific tests for phenolic compounds and other compounds, e.g., ascorbic acid, may also influence the result [32]. Conducted correlation analysis also shows small amount of statistically significant correlations between assays and identified compounds, which may indicate that obtained extracts are complex mixture of compounds and should be evaluated further. Strong positive correlations were observed between DPPH and total phenol content, as well as total flavonoid content and FRAP assay, but they were not statistically significant. In the study of different types of honey derived from Mount Olympus in Greece, authors observed similar effect, which can be explained by fact, that the antioxidant properties depend not on the quantity, but mainly on the chemical composition of polyphenols and other factors may affect results of conducted assays, such as the concentration of mineral contents, organic acids, amino acids [33]. Lack of correlation between conducted antiradical ant antioxidant assays can be explained by various complex mechanisms of neutralizing free radicals, e.g., Brand-Williams et al. and Tagashira et al. found no correlation between the content of phenolic acids and their antioxidant activity [34,35].

All extracts were characterized by antimicrobial properties in the screening test, where the strongest were pine and spruce extracts. In study on antimicrobial activity of aqueous extracts of *Juniperus phoenicea*, *Pistacia atlantica* and *Oudneya africana* tested extracts of *Juniperus* exhibited similar bacterial inhibition against *Listeria ivanovii* RBL30, *Listeria innocua* RBL29 and *Listeria monocytogenes* LSD530 [36]. In study of essential oils from *Pinus halepensis* Mill., essential oils showed strong inhibiting activity against *L. monocytogenes* and *Klebsiella pneumoniae,* There is much evidence in the literature for the antimicrobial effect of pine and juniper, however, tested properties are usually found in concentrated essential oils or ethanol extracts [37]. Pearson’s correlation test revealed strong positive correlation between sinapic acid content and inhibition of *Salmonella enteritidis* ATCC 13,076. In the literature there are mentions about antimicrobial effects of phenolic compounds on *Salmonella* bacteria, notwithstanding sinapic acid is not widely recognized as antimicrobial compound, but many phenolic compounds may exhibit significant antibacterial activity, however mechanisms of antibacterial action of phenolic compounds are not yet fully deciphered [38,39]. Correlation analysis showed also statistically significant negative correlations between inhibition of *Staphylococcus aureus* ATCC 25,923 and ferulic acid content, *Aspergillus* spp. Inhibition and FCR assay. Different authors also observed negative correlations or lack of thereof between FCR assay and antimicrobial properties, therefore synergistic or antagonistic interactions between compounds cannot be fully explained by a simple linear relation [33,40].

## 4. Materials and Methods

### 4.1. Materials

The study material consisted of shoots samples from six different coniferous trees: *Picea abies* L. (PA), *Larix decidua* Mill; (LD), *Pinus sylvestris* L. (PS), *Pseudotsuga menziesii* (PM) and *Juniperus communis* L. (JC) collected in 2019 from the arboretum in Zielonka (Poland, 17°06′33″ E, 52°06′33″ N), a part of the Forest Experimental Department of Poznan University of Life Sciences. The shoots were collected in August and subjected to natural air-drying at 21 °C for 72 h. The dried needles were sampled from three different shoots, crushed in a Grindomix GM 200 (Retsch GmbH, Haan, Germany) for 15 s at a rate of 500 rpm at 21 °C to a particle size of 0.5–0.9 mm.

### 4.2. Methods

#### 4.2.1. Extraction

Water extracts were obtained by mixing 5 g of raw material with 150 mL of distilled water. The samples were shaken in a water bath for 30 min at 80 °C at constant amplitude. Extracts were decanted and filtered using Whatman No. 4 paper three times. The obtained extracts were stored at −21 °C for no more than two weeks before further analyses. Each measurement and analysis for each extract were conducted in triplicate.

#### 4.2.2. Commodity Assessment, Color and Osmolality of Extracts

Shoots were subjected to commodity assessment in terms of characteristics and shape in the laboratory in daylight. Needle dimensions were determined by measuring the length of 100 needles of each species using a caliper and calculating the average. Dry weight of the shoots was determined using Sartorius MA 30 (Sartorius AG, Goettingen, Germany), where 1 g of the sample was incinerated at 130 °C for 30 min.

Colors of leaf extracts were measured. Color measurement was run in the L* a* b* CEN unit system using a CM-5 spectrometer (Konica Minolta, Tokyo, Japan) according to the methodology described by the device producer. D 65 was applied as a source of light and color temperature was 6504 K. The observation angle of the standard colorimetric observer was 10°. Measurements for each sample was repeated five times. Instrument calibration was performed using the black pattern.

#### 4.2.3. HPLC Determination of Phenolic Acids and Flavonols

Extracts were evaporated to dryness in a stream of nitrogen. Next they were placed in sealed 17-mL culture test tubes, where first alkaline and then acid hydrolysis was run. In order to run alkaline hydrolysis 1 mL distilled water and 4 mL 2-M aqueous sodium hydroxide was added to test tubes. Tightly sealed test tubes was heated in a water bath at 95 °C for 30 min. After cooling (approx. 20 min) test tubes was neutralized with 2 mL 6-M aqueous hydrochloric acid solution (pH = 2). Next, samples were cooled in water with ice. Flavonoids were extracted from the inorganic phase using diethyl ether (2 × 2 mL). Formed ether extracts were continuously transferred to 8-mL vials. Next acid hydrolysis was run. For this purpose, the aqueous phase was supplemented with 3 mL 6 M aqueous hydrochloric acid solution. Tightly sealed test tubes were heated in a water bath at 95 °C for 30 min. After being cooled in water with ice the samples were extracted with diethyl ether (2 × 2 mL). Produced ether extracts were continuously transferred to 8-mL vials, after which they were evaporated to dryness in a stream of nitrogen. Prior to analyses samples were dissolved in 1 mL methanol. Phenolic compounds analysis was performed using an Acquity H class UPLC system equipped with a Acquity PDA detector (Waters Corp, Milford, MA, USA). Chromatographic separation was performed on an Acquity UPLC^®^ BEH C_18_ column (100 mm × 2.1 mm, particle size—1.7 μm) (Watersy, Dublin, Ireland). Elution was carried out in a gradient using the following mobile phase composition: A: acetonitrile with 0.1% formic acid, B: 1% aqueous formic acid mixture (pH = 2). Concentrations of phenolic compounds were determined using an internal standard at wavelengths λ = 320 nm and 280 nm and finally expressed as mg/100 g dw of sample. Compounds were identified by comparing the retention time of the analyzed peak with the retention time of the standard and by adding a specific amount of the standard to the analyzed samples and repeated analysis. Detection level was 1 μg/g. Retention times for phenolic acids were as follows: gallic acid—4.85 min, *p*-coumaric acid—8.06 min, 2,5-dihydroxybenzoic acid—9.55 min, 4-hydroxybenzoic acid—9.89 min., chlorogenic acid—12.00 min, caffeic acid—15.20 min, syringic acid—15.60 min, vanillic acid—16.80 min, sinapic acid—17.10 min, ferulic acid—17.50 min, salicylic acid—17.85 min., *t-* cinnamic acid—19.50 min Retention times for flavonoids were as follows: vitexin—1.10 min, apigenin—8.00 min, kaempferol—11.00 min., luteolin—16.90 min., quercetin—17.00 min, naringenin—17.50 min, rutin—17.90 min [30].

#### 4.2.4. Folin-Ciocâlteu Reagent Assay

Reducing capacity of the obtained extracts was determined using the Folin-Ciocâlteu reagent (FCR) and the method of Kobus-Cisowska et al. with minor modifications [41]. Reducing capacity was expressed as mg of gallic acid (Sigma-Aldrich, Steinheim, Germany) equivalents (GAE) per 1 g (mg/1 g) of dry mass. The standard curve in the range of 0–500 mg/mL of gallic acid was used.

#### 4.2.5. Total Flavonoid Content

Total flavonoid content was determined using a procedure described by Meda et al. [42]. Total flavonoid content was determined using a standard curve with quercetin (Sigma-Aldrich, Germany) concentration in range of 1–100 µg/mL. The mean of three readings was used and expressed as mg of quercetin equivalents QE/1 g raw material.

#### 4.2.6. Ferric Reducing Antioxidant Power Assay

The antioxidant properties of water extracts were determined using the ferric reducing/antioxidant power assay (FRAP method) according to the procedure described by O’Sullivan et al. [43]. The calibration curve was constructed using FeSO_4_·7H_2_O in concentrations of 100–1000 µM. Samples were incubated for 30 min and the absorbance was measured at 593 nm (Metertech SP880, Metertech Inc., Taipei Taiwan). Data were expressed as µM FeSO_4_/g dry mass.

#### 4.2.7. DPPH Radical Scavenging Activity

DPPH inhibition capacity was investigated according to the procedure described by Szczepaniak et al. [44]. The calibration curve was prepared using Trolox standard solution in concentrations of 100–1000 µM. The decrease in DPPH absorbance was measured at 517 nm according to the blank. Inhibition capacity of DPPH was expressed as μM Trolox/g dw.

#### 4.2.8. Antimicrobial Screening

Indicator microorganisms *Klebsiella pneumoniae* ATCC 31,488, *Salmonella enteritidis* ATCC 13,076, *Pseudomonas aeruginosa* ATCC 27853, *Acinetobacter baumannii* ATCC 19606, *Enterococcus* faecium ATCC 27,270, *Staphylococcus aureus* ATCC 25,923, *Lactobacillus fermentum* ATCC 14,932, *Clostridium butyricum* ATCC 860, *Listeria monocytogenes* ATCC 19,115, *Bacillus coagulans*, GBI-30, 6086, *Candida utilis* ATCC 9950, *Aspergillus* spp. and *Fusarium* spp. were transferred to test tubes containing Mueller-Hinton (for bacteria), yeast extract sucrose (for yeast), potato dextrose (for mold) medium. They were cultured at 37 °C for 24 h. Next, liquefied agar medium was inoculated with 10% (*v/v*) 24-h indicator culture and poured into Petri dishes to obtain a distinct confluent layer. After solidification of the broth medium inoculated with the indicator microorganisms, wells were made with a cork borer. Each well was supplemented with 150 µL of aqueous extracts of coniferous shoots. Next, the diameters of the growth inhibition or reduction zone of indicator microorganisms were measured. The inhibition of the growth of the indicator microorganism was manifested by complete lightening around the place where the liquid extract or slime was transferred. It indicated bactericidal activity of the bacterial strain. Bacteriostatic properties were determined by measuring the diameter of the growth inhibition zone (indicator strain growth limitation).

#### 4.2.9. Statistical Analysis

All assays were conducted in triplicate and the results were expressed as mean ± SD. One-way ANOVA was used to analyze statistical differences between different extracts in terms of phenolic compound contents and different antioxidant assays with the least significant difference (LSD). A *p*-value less than 0.05 was considered to be statistically significant. Correlation analysis was conducted using Pearson parametric correlation test. Statistical analyses were calculated using Statistica 13.3 software (TIBCO, Palo Alto, CA, USA) and RStudio (RStudio PBC, 1.3.1056, Boston, MA, USA).

## 5. Conclusions

The present work has shown that aqueous extracts obtained from sample shoots of *Picea abies* L., *Larix decidua* Mill, *Pinus sylvestris* L., *Pseudotsuga menziesii* and *Juniperus communis* L. exhibit antioxidant and antimicrobial properties in vitro. They are characterized by a high content of phenolic compounds—among others—4-hydroxybenzoic acid, caffeic acid, ferulic acid and chlorogenic acid, i.e., compounds considered to have a wide spectrum of pro-health effects. Shoots of the studied conifers differ from each other in terms of physical parameters as well as antioxidant and antimicrobial properties; however, the results of the research allow to conclude that after further necessary analyses, i.e., cytotoxicity and sensory tests, these raw materials could potentially be used as components of functional food with programmed health-promoting properties, as antioxidant ingredients and those extending the shelf life of the products.

## Figures and Tables

**Figure 1 molecules-25-03527-f001:**
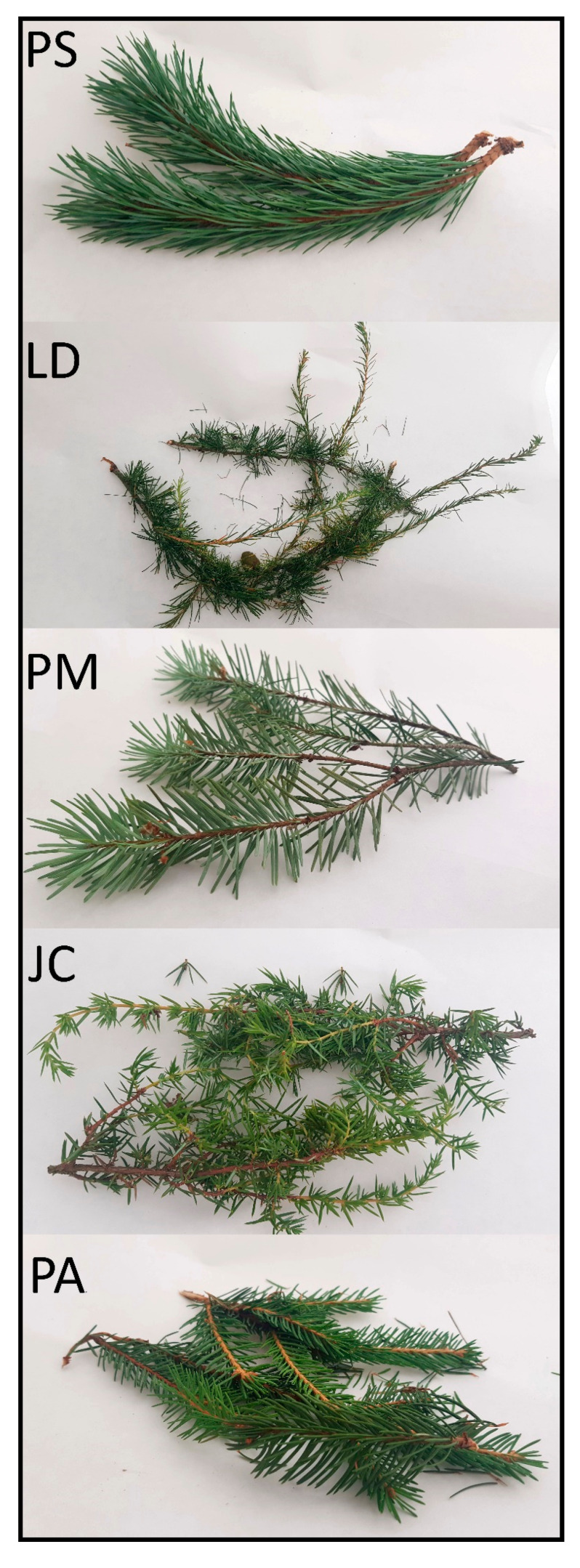
Fresh conifer shoots. Abbreviations: *Picea abies* L. (PA), *Larix decidua* Mill (LD), *Pinus sylvestris* L. (PS), *Pseudotsuga menziesii* (PM) and *Juniperus* communis L. (JC).

**Figure 2 molecules-25-03527-f002:**
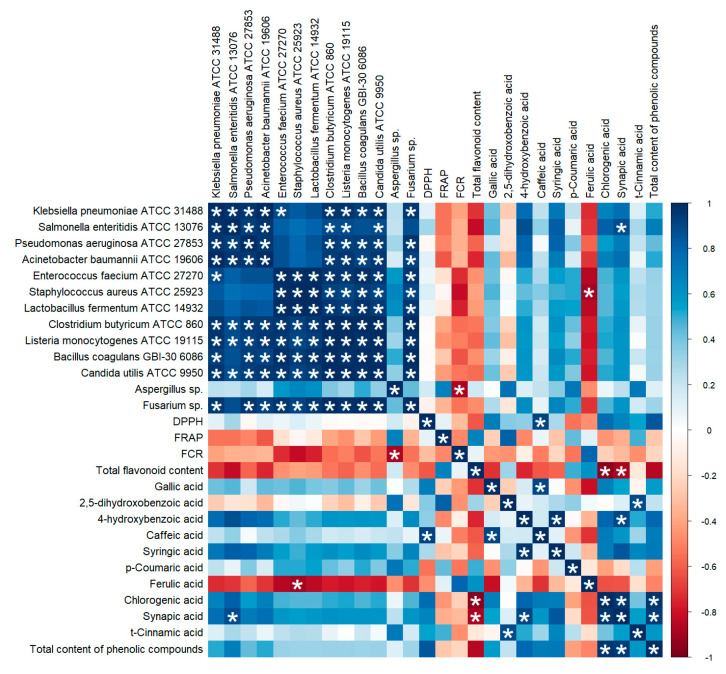
Correlation matrix. The intensity of the color is proportional to the correlation coefficient; white asterisk indicates a significantly statistical correlation (*p* < 0.05).

**Table 1 molecules-25-03527-t001:** Commodity assessment of tested coniferous shoots.

Parameter		*Pseudotsuga menziesii*	*Juniperus communis* L.	*Larix decidua* Mill	*Pinus sylvestris* L.	*Picea abies* L.
**Needle Shape**		Flattened, cylindrical	Wide, flattened diamonds	Thin, filamentous	filamentous	Elongated diamond
**Needle Length (mm)**		23.27 ^a^ ± 2.99	8.74 ^b^ ± 3.08	25.31 ^c^ ± 5.26	54.84 ^d^ ± 9.91	14.32 ^e^ ± 2.91
**Color**	L*	33.74 ^a^ ± 2.50	38.83 ^b^ ± 0.35	29.31 ^c^ ± 0.14	30.33 ^c^ ± 0.78	26.09 ^d^ ± 0.56
	a*	−0.18 ^a^ ± 0.99	−5.25 ^b^ ± 0.20	2.14 ^c^ ± 0.22	−1.14 ^a^ ± 1.19	8.26 ^d^ ± 0.91
	b*	13.43 ^a^ ± 0.18	24.48 ^b^ ± 0.21	15.28 ^c^ ± 0.71	10.73 ^d^ ± 0.23	14.82 ^c^ ± 0.36
**Dry Weight (%)**		13.25 ^a^ ± 0.35	10.95 ^b^ ± 0.35	16.16 ^c^ ± 0.49	32.22 ^d^ ± 0.49	13.98 ^a^ ± 0.49
**Extract Osmolality** **(mOsm/kg H_2_O)**		0.04 ^a^ ± 0.00	0.04 ^b^ ± 0.00	0.05 ^c^ ± 0.00	0.02 ^d^ ± 0.00	0.05 ^e^ ± 0.00
**Freezing Temperature (°C)**		−0.08 ± 0.00	−0.07 ± 0.00	−0.09 ± 0.00	−0.04 ± 0.00	−0.09 ± 0.00

Results are mean values of three determinations ± standard deviation. Values sharing the same letter in a line are not significantly different (*p* ≤ 0.05).

**Table 2 molecules-25-03527-t002:** HPLC analysis of phenolic compounds in tested conifer shoot extracts.

Phenolic Acid (µg/g dw)	LD	JC	PM	PS	PA
**Gallic Acid**	10.86 ^b^ ± 0.48	994.72 ^c^ ± 47.49	57.04 ^d^ ± 1.27	208.38 ^e^ ± 069	695.88 ^f^ ± 5.29
**2,5-Dihydroxobenzoic Acid**	130.11 ^b^ ± 6.8	25.55 ^c^ ± 0.15	7.18 ^c^ ± 0.16	16.63 ^c^ ± 0.54	62.43 ^d^ ± 1.95
**4-Hydroxybenzoic Acid**	622.99 ^b^ ± 24.61	22.96 ^b^ ± 1.44	1148.62 ^b^ ± 23.72	1084.92 ^b^ ± 39.04	4014.44 ^c^ ± 58.25
**Caffeic Acid**	2994.35 ^b^ ± 104.77	5999.36 ^c^ ± 156.04	1499.61 ^d^ ± 36.84	1502.03 ^d^ ± 52.53	5094.84 ^e^ ± 228.14
**Syringic Acid**	139.15 ^b^ ± 3.89	50.12 ^c^ ± 1.79	113.97 ^d^ ± 4.2	145.44 ^b^ ± 3.28	301.96 ^e^ ± 9.55
***p*-Coumaric Acid**	298.03 ^b^ ± 6.58	82.49 ^c^ ± 4.26	68.75 ^c^ ± 3.39	387.89 ^d^ ± 15.83	168.58 ^a^ ± 10.89
**Ferulic Acid**	3708.83 ^b^ ± 127.71	1379.03 ^c^ ± 14.44	5002.20 ^d^ ± 212.87	2088.89 ^e^ ± 56.89	1129.85 ^f^ ± 31.1
**Chlorogenic Acid**	501.97 ^b^ ± 22.84	2093.81 ^c^ ± 34.93	984.09 ^d^ ± 23.28	518.25 ^b^ ± 4.90	4534.29 ^e^ ± 227.15
**Sinapic Acid**	43.61 ^b^ ± 1.64	214.18 ^c^ ± 3.68	6.86 ^a^ ± 0.18	54.09 ^b^ ± 2.06	1172.00 ^d^ ± 24.37
***t*-Cinnamic Acid**	819.74 ^b^ ± 29.33	127.53 ^a^ ± 1.91	55.86 ^c^ ± 2.63	111.44 ^a^ ± 3.4	781.83 ^d^ ± 40.05
**Vanillic Acid**	0.33 ^b^ ± 0.00	0.47 ^c^ ± 0.01	0.95 ^d^ ± 0.02	0.46 ^c^ ± 0.01	1.56 ^e^ ± 0.01
**Salicylic acid**	0.36 ^b^ ± 0.00	0.75 ^c^ ± 0.01	1.04 ^d^ ± 0.01	0.36 ^b^ ± 0.00	0.34 ^a^ ± 0.01
**Naringenin**	1.00 ^b^ ± 0.02	1.03 ^c^ ± 0.08	1.04 ^b^ ± 0.02	1.59 ^d^ ± 0.02	1.42 ^e^ ± 0.06
**Vitexin**	0.53 ^b^ ± 0.00	1.11 ^c^ ± 0.02	0.78 ^d^ ± 0.02	0.61 ^e^ ± 0.01	0.30 ^f^ ± 0.00
**Rutin**	0.52 ^b^ ± 0.01	1.14 ^c^ ± 0.02	0.73 ^d^ ± 0.03	0.63 ^e^ ± 0.02	0.31 ^f^ ± 0.01
**Quercetin**	0.63 ^b^ ± 0.00	0.64 ^b^ ± 0.01	1.38 ^c^ ± 0.06	0.98 ^d^ ± 0.03	1.24 ^e^ ± 0.04
**Apigenin**	0.62 ^b^ ± 0.02	0.30 ^a^ ± 0.01	0.30 ^a^ ± 0.00	0.30 ^a^ ± 0.01	0.31 ^a^ ± 0.01
**Kaempferol**	0.30 ^a^ ± 0.01	0.31 ^a^ ± 0.00	0.33 ^b^ ± 0.01	0.38 ^c^ ± 0.01	0.36 ^d^ ± 0.01
**Luteolin**	0.30 ^a^ ± 0.01	0.30 ^a^ ± 0.01	0.31 ^a^ ± 0.01	0.30 ^a^ ± 0.01	0.30 ^a^ ± 0.01
**Total Content**	9274.23	10,995.8	8951.04	6123.57	13,947.80

Results are mean values of three determinations ± standard deviation. Values sharing the same letter in a line are not significantly different (*p* ≤ 0.05). Abbreviations: *Picea abies* L. (PA), *Larix decidua* Mill (LD), *Pinus sylvestris* L. (PS), *Pseudotsuga menziesii* (PM) and *Juniperus communis* L. (JC).

**Table 3 molecules-25-03527-t003:** Radical scavenging and antioxidant properties of tested conifers shoots using spectrophotometric methods.

Species	DPPH (μM Trolox/g dw)	FRAP (μM FeSO_4_/g dw)	FCR (mg GAE/g dw)	Total Flavonoid Content (mg QE/g dw)
***Picea abies*** **L.**	404.18 ^a^ ± 10.15	15.37 ^a^ ± 2.55	13.30 ^a^ ± 0.55	3.54 ^a^ ± 0.19
***Pinus sylvestris*** **L.**	200.94 ^b^ ± 23.47	42.76 ^b^ ± 5.7	0.86 ^b^ ± 0.09	8.29 ^b^ ± 0.94
***Pseudotsuga menziesii***	269.55 ^c^ ± 6.31	5.43 ^a^ ± 1.58	14.53 ^c^ ± 0.64	7.46 ^c^ ± 0.27
***Juniperus communis*** **L.**	384.30 ^a^ ± 10.88	62.88 ^c^ ± 0.36	8.25 ^d^ ± 1.01	6.34 ^d^ ± 0.09
***Larix decidua*** **Mill**	326.93 ^d^ ± 21.21	147.94 ^d^ ± 21.86	14.83 ^c^ ± 0.30	9.90 ^e^ ± 0.12

Results are mean values of three determinations ± standard deviation. Values sharing the same letter in a line are not significantly different (*p* ≤ 0.05). Abbreviations: Folin-Ciocâlteu reagent (FCR), the ferric reducing ability of plasma (FRAP), 2,2-diphenyl-1-picrylhydrazyl. (DPPH), gallic acid equivalents (GAE), quercetin equivalents (QE).

**Table 4 molecules-25-03527-t004:** Antimicrobial properties of tested conifer shoots.

Microorganism		PM	JC	LD	PS	PA
		Growth Inhibition Zone (mm)				
**Gram-Negative Bacteria**						
1	*Klebsiella pneumoniae* ATCC 31,488	8 ^a^ ± 2	12 ^b^ ± 2	5 ^c^ ± 1	22 ^d^ ± 2	28 ^e^ ± 3
2	*Salmonella enteritidis* ATCC 13076	6 ^a^ ± 1	9 ^b^ ± 2	3 ^c^ ± 1	16 ^d^ ± 2	29 ^e^ ± 3
3	*Pseudomonas aeruginosa* ATCC 27853	11 ^a^ ± 2	10 ^a^ ± 2	9 ^b^ ± 2	27 ^c^ ± 3	32 ^d^ ± 2
4	*Acinetobacter baumannii* ATCC 19606	8 ^a^ ± 1	11 ^b^ ± 2	3 ^c^ ± 1	20 ^d^ ± 2	26 ^e^ ± 3
**Gram-Positive Bacteria**						
5	*Enterococcus faecium* ATCC 27270	4 ^a^ ± 1	12 ^b^ ± 2	9 ^c^ ± 1	18 ^d^ ± 2	19 ^d^ ± 2
6	*Staphylococcus aureus* ATCC 25923	2 ^a^ ± 0	15 ^b^ ± 3	11 ^c^ ± 2	19 ^d^ ± 2	22 ^e^ ± 2
7	*Lactobacillus fermentum* ATCC 14932	3 ^a^ ± 1	17 ^b^ ± 3	13 ^c^ ± 2	13 ^c^ ± 2	29 ^d^ ± 3
8	*Clostridium butyricum* ATCC 860	9 ^a^ ± 2	16 ^b^ ± 2	10 ^c^ ± 1	17 ^d^ ± 2	28 ^e^ ± 3
9	*Listeria monocytogenes* ATCC 19,115	8 ^a^ ± 2	15 ^b^ ± 2	7 ^a^ ± 1	19 ^c^ ± 2	25 ^d^ ± 3
10	*Bacillus coagulans* GBI-30, 6086	7 ^a^ ± 2	12 ^b^ ± 1	10 ^c^ ± 2	19 ^d^ ± 2	21 ^e^ ± 3
**Fungi**						
11	*Candida utilis* ATCC 9950	3 ^a^ ± 1	5 ^b^ ± 1	3 ^a^ ± 1	6 ^c^ ± 1	8 ^d^ ± 2
12	*Aspergillus* sp.	4 ^a^ ± 1	4 ^a^ ± 1	6 ^b^ ± 1	5 ^c^ ± 1	5 ^c^ ± 1
13	*Fusarium* sp.	2 ^a^ ± 0	5 ^b^ ± 1	2 ^a^ ± 0	5 ^a^ ± 1	9 ^b^ ± 2

Results are mean values of three determinations ± standard deviation. Values sharing the same letter in a line are not significantly different (*p* ≤ 0.05). Abbreviations: *Picea abies* L. (PA), *Larix decidua* Mill (LD), *Pinus sylvestris* L. (PS), *Pseudotsuga menziesii* (PM) and *Juniperus communis* L. (JC).

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
