# Peer review of "Identification of Polyphenols from Coniferous Shoots as Natural Antioxidants and Antimicrobial Compounds"

_molecules, 2020, doi:10.3390/molecules25153527_

Round 1
Reviewer 1 Report
Dears Editor and Authors
The manuscript presented is a very valuable and intriguing working report, focused on the use of alternative strategies against Gram-positive bacteria, more precisely due to the secondary metabolites present in conifer needles. In this context, I believe that the manuscript submitted is welcome.
The abstract has all the components present, it is well written concisely, clearly. The data is appropriate and the subject is original and important.
I have a few comments to make:
row 80 It must write italic in vitro
Key words
to be modified antioxidant properties, antimicrobial properties, in antioxidant and antimicrobial properties
Introduction: is well written, concise, it used relevant, well documented with relevant and recent literature regarding the importance constituents of coniferous shoots as well as the role of its secondary metabolites on health.
Results: are logically presented in a concise, easy to follow manner, also using the figures, which are well organized. The result are summarized important observation and well written.
I have a few comments to make:
row 80 to be modified The highest dry weight content in The top-level of dry weight
row 86 to mention the names of the coniferous species in the figure
line 98 I do not understand what 4.3 represents
row 119 The name of the coniferous species used without abbreviation should be entered in the table
line 129 The species P. aeruginosa and S. aureus must be written in italics
row 130 L. fermentum written in italics
Discussion: contain precise comments based on found results. These emphasize in conformity with the results obtained the possible role of metabolites, isolated and quantified from coniferous shoots against some Gram positive and Gram negative bacteria as well as fungi. Throughout this section, many references are being used, in order to support the given comments.
Material and methods
I did not understand what the authors wanted to convey
The origin of the biological material used should be written both for the species of shoots and for Gram-positive, negative bacteria and fungi.
I believe that the biochemical methods used to determine the secondary metabolites both quantitatively and qualitatively should be mentioned.
My conclusion is that these results are very valuable, interesting also can also open new perspectives in used of extracts from some coniferous shoots because of their antioxidant and antimicrobial effects as well as also due to its low toxicity and rapid bacterial killing. In this way, this kind of research may be also promising for economical point of view, by using of metabolites secondary from coniferous shoots in pharmaceutical and alimentary industries.
Author Response
Dear reviewer,
Thank you for your thorough review. All comments and recommendations have been included in the revised version of the manuscript. Changes to the manuscript were made in the content of the publication. The comments listed in the reviews are referenced in the table below.
We hope that the current state of the manuscript meets the standards of the journal and that publication will be possible.
Best regards,
Marcin Dziedziński
Comments and responses:
- row 80 It must write italic in vitro
Ad 1. It has been fixed
- Key words to be modified antioxidant properties, antimicrobial properties, in antioxidant and antimicrobial properties
Ad 2. It has been fixed
- row 80 to be modified The highest dry weight content in The top-level of dry weight
Ad 3. It has been fixed
- row 86 to mention the names of the coniferous species in the figure
Ad 4. It has been added.
- line 98 I do not understand what 4.3 represents
Ad 5. It has been fixed
- row 119 The name of the coniferous species used without abbreviation should be entered in the table
Ad 6. It has been fixed
- line 129 The species P. aeruginosa and S. aureus must be written in italics
Ad 7. It has been fixed
- row 130 L. fermentum written in italics
Ad 8. It has been fixed
- Material and methods
I did not understand what the authors wanted to convey
The origin of the biological material used should be written both for the species of shoots and for Gram-positive, negative bacteria and fungi.
I believe that the biochemical methods used to determine the secondary metabolites both quantitatively and qualitatively should be mentioned.
Ad 9. We revised the methodology.
Reviewer 2 Report
Dear authors, your work is quite interesting.
I have considered the materials and methods section of your interesting study and here are some comments to consider.
1) Please add in all sections the range of concentrations you used for your standards solutions employed for the construction of your calibration curve
2) in section that you describe the hplc analysis, please briefly describe the alkaline and acid hydrolysis steps you did, since neither in the cited reference this information is shown. Have in mind that alkaline and acid hydrolysis lead to the formation of new phenols from initial complex molecules. So the phenolic profile you mention is related to this extracts and not the aqueous ones for which the rest information is provided. So, this is something that needs to be mentioned in your discussion section.
3) moreover as already mentioned in previous review comments your discussion section could be probably enriched by addition of comments on correlations od data and relation with other published data even in different materials that may possess similar content and bioactive performance.
4) Lastly as stressed in the 1st review provided by me, you need to stress that these findings concern data from the analysis of one case sample for each specie. Post and preharvest, biotic and abiotic parameters may significantly affect the bioactive content and composition of plant materials.
Author Response
Dear reviewer,
Thank you for your thorough review. All comments and recommendations have been included in the revised version of the manuscript. Changes to the manuscript were made in the content of the publication. The comments listed in the reviews are referenced below.
We hope that the current state of the manuscript meets the standards of the journal and that publication will be possible.
Best regards,
Marcin Dziedziński
Comments and responses:
- Please add in all sections the range of concentrations you used for your standards solutions employed for the construction of your calibration curve
Ad 1. Ranges of concentrations have been added.
- in section that you describe the hplc analysis, please briefly describe the alkaline and acid hydrolysis steps you did, since neither in the cited reference this information is shown. Have in mind that alkaline and acid hydrolysis lead to the formation of new phenols from initial complex molecules. So the phenolic profile you mention is related to this extracts and not the aqueous ones for which the rest information is provided. So, this is something that needs to be mentioned in your discussion section.
Ad 2. I agree with the reviewer, this methodology contains an error resulting from its application to plant material. Unnecessary items have been removed.
- moreover as already mentioned in previous review comments your discussion section could be probably enriched by addition of comments on correlations od data and relation with other published data even in different materials that may possess similar content and bioactive performance.
Ad 3. We extended discussion section and conducted correlation analysis.
4) Lastly as stressed in the 1st review provided by me, you need to stress that these findings concern data from the analysis of one case sample for each specie. Post and preharvest, biotic and abiotic parameters may significantly affect the bioactive content and composition of plant materials.
Ad 4. We've added explanatory sentences in abstract, introduction and methodology emphasizing that these are individual samples.
Reviewer 3 Report
The work deals with an interesting and new argument, such as the evaluation of the antioxidant potential of coniferous shoots. However, it presents some important issues to be clarified.
General comments
Introduction. This section is sufficient.
R&D. Results are too sketchy described and should be deeper analyzed. Moreover, both qualitative and quantitative data on phenol content has not been compared to previous literature data. This is a negative comment.
M&M. This section is completely absent. This fact allows to give a very negative comment for the possible publication of the paper.
Conclusions. This section is not sufficient, given the previous comments on R&D and M&M.
The summary of the comments give the chance to consider the proposed work not suitable for publication in Molecules, with no chance to be edited for an amelioration.
Specific comments
-line 24: if possible, gave the name of some relevant phenol compound found in the analyzed extracts of coniferous, since phenols is too generic.
- line 26: uniform the decimal points in the given values (just as an example, all data have two decimal points, 13947.80 should be given).
- lines 33-35: the sentence is not clear. PA and PS had the highest activity against gram-negative bacteria, low for gram-positive and fungi? What species resulted high in these latter cases?
- lines 50-51: it should be indicated that the terpenoid compounds are part of the resins, belonging to the insoluble fraction.
- lines 68-69: the abbreviations of analysed samples should be better if placed here, instead of in the Abstract.
- line 73-74: please, add the measure units to the lenght of selected shoots.
- line 86, Table 1: please, uniform the decimal points, where necessary. Add the name of the species besides the abbreviations. Figure and Tables have to be the most possible self-explaining.
- line 90: write “Pseudotsuga menziesii” in Italic, it is a scientific name. This also in other Tables.
- lines 95-96: from my vision of Table 2, I see that, among the relevant phenol compounds found in coniferous shoots, there is also caffeic acid, ranging from 1500 to 5999 microg/g d.w.. Why it was not cited among the relevant compounds?
- line 100, Table 2: please, check the formatting of column data for PA sample. It is not clear. Moreover, check the correct writing of P-coumaric and T-cinnamic. (p-coumaric and t-cinnamic?).
- line 118, Table 3: some corrections are needed. Write correctly FeSO4 and explain what means “Q” in the determination of total flavonoid content.
- lines 93-123: the given results are surely very interesting and, to my opinion should be enriched by a deeper data analysis, for example by giving some correlation indexes among the assayed parameters, in order to establish the role of phenols in the parameters of antioxidant capacity and if the different assays of reducing power (FCR and FRAP) are related or not with the DPPH assay and/or with the total flavonoid assay. Moreover, very interesting should be also the study of the possible relationship amomg the antioxidant parameters and the phenol content with the antimicrobial assays. This could strongly enhance the scientific value of the work.
- lines 128-129: write P. aeruginosa in Italic. The same for following microbial species in the paragraph.
- lines 131-132: this sentence is also in the Abstract, but it is not clear. Please, rewrite it.
- line 133, Table 4: write correctly “Candida utilis”.
- line 149: we are not sure if the antioxidant activity is exclusively determined by phenols and not by other chemical species, not analyzed in the present work. Should it be used the term “... which significantly contributed to ...”?
Author Response
Dear reviewer,
Thank you for your thorough review. All comments and recommendations have been included in the revised version of the manuscript. Changes to the manuscript were made in the content of the publication. The comments listed in the reviews are referenced below.
We hope that the current state of the manuscript meets the standards of the journal and that publication will be possible.
Best regards,
Marcin Dziedziński
Comments and responses:
- R&D. Results are too sketchy described and should be deeper analyzed. Moreover, both qualitative and quantitative data on phenol content has not been compared to previous literature data. This is a negative comment.
Ad 1. We conducted correlation analysis and enriched discussion.
- M&M. This section is completely absent. This fact allows to give a very negative comment for the possible publication of the paper.
Ad 2. It has been resolved.
- This section is not sufficient, given the previous comments on R&D and M&M.
- The summary of the comments give the chance to consider the proposed work not suitable for publication in Molecules, with no chance to be edited for an amelioration.
Specific comments
- -line 24: if possible, gave the name of some relevant phenol compound found in the analyzed extracts of coniferous, since phenols is too generic.
Ad 5. Names of significant phenols have been added
- - line 26: uniform the decimal points in the given values (just as an example, all data have two decimal points, 13947.80 should be given).
Ad 6. We fixed that issue in the whole manuscript.
- - lines 33-35: the sentence is not clear. PA and PS had the highest activity against gram-negative bacteria, low for gram-positive and fungi? What species resulted high in these latter cases?
Ad 7. This sentence has been revised.
- - lines 50-51: it should be indicated that the terpenoid compounds are part of the resins, belonging to the insoluble fraction.
Ad 8. This information has been added.
- - lines 68-69: the abbreviations of analysed samples should be better if placed here, instead of in the Abstract.
Ad 9. We placed the abbreviations in mentioned lines.
- - line 73-74: please, add the measure units to the lenght of selected shoots.
Ad 10. The unit (mm) is included.
- - line 86, Table 1: please, uniform the decimal points, where necessary. Add the name of the species besides the abbreviations. Figure and Tables have to be the most possible self-explaining.
Ad 11. It has been fixed accordingly.
- - line 90: write “Pseudotsuga menziesii” in Italic, it is a scientific name. This also in other Tables.
Ad 12. It has been corrected.
- - lines 95-96: from my vision of Table 2, I see that, among the relevant phenol compounds found in coniferous shoots, there is also caffeic acid, ranging from 1500 to 5999 microg/g d.w.. Why it was not cited among the relevant compounds?
Ad 13. It has been omitted by mistake, we corrected that.
- line 100, Table 2: please, check the formatting of column data for PA sample. It is not clear. Moreover, check the correct writing of P-coumaric and T-cinnamic. (p-coumaric and t-cinnamic?).
Ad 14. It has been corrected accordingly.
- - line 118, Table 3: some corrections are needed. Write correctly FeSO4 and explain what means “Q” in the determination of total flavonoid content.
Ad 15. It has been corrected and abbreviations explained.
- - lines 93-123: the given results are surely very interesting and, to my opinion should be enriched by a deeper data analysis, for example by giving some correlation indexes among the assayed parameters, in order to establish the role of phenols in the parameters of antioxidant capacity and if the different assays of reducing power (FCR and FRAP) are related or not with the DPPH assay and/or with the total flavonoid assay. Moreover, very interesting should be also the study of the possible relationship amomg the antioxidant parameters and the phenol content with the antimicrobial assays. This could strongly enhance the scientific value of the work.
Ad 16. We conducted correlation analysis as Figure 2.
- - lines 128-129: write P. aeruginosa in Italic. The same for following microbial species in the paragraph.
Ad 17. It has been corrected.
- - lines 131-132: this sentence is also in the Abstract, but it is not clear. Please, rewrite it.
Ad 18. We revised that sentence.
- - line 133, Table 4: write correctly “Candida utilis”.
Ad 19. It has been corrected.
- - line 149: we are not sure if the antioxidant activity is exclusively determined by phenols and not by other chemical species, not analyzed in the present work. Should it be used the term “... which significantly contributed to ...”?
Ad 20. We corrected that sentence.
Round 2
Reviewer 2 Report
Nice corrections throughout.
Your title has a grammar spelling error and has to be corrected.
Author Response
Dear Reviewer,
We corrected the title to "Identification of polyphenols from coniferous shoots as natural antioxidants and antimicrobial compounds".
Thank you for your comments and suggestions, we hope the manuscript meets the requirements for publication in the journal.
Sincerely, Marcin Dziedzinski
Reviewer 3 Report
The manuscript has been revised by the Authors, after the numerous comments made by Referees.
Now it seems ameliorated. As a general comment, the work still lacks of a deep discussion and literature comparison of the obtained results and of a clear report of the used analytical methodologies. Hence, unfortunately, it could be considered for publication only after several further “major recommendations”, as follows.
Line 53: sorry, my mistake in the suggested correction: “… belongs to water-insoluble fraction.”
Line 70: correct “conifers”.
Line 106, Table 2: correct “kaempferol”.
Line 197: please, add the measure units of the given values 60-6390.
Line 201: please correct to “μmoles/g”, or “mM”, optionally.
Lines 217-219: also if the analyzed solutions are complex in their phenolic composition, a deeper explanation enforced by comparison with existing literature of the calculated correlations with antioxidant indexes must be given. Very interesting are the found positive correlations between caffeic acid and gallic acid and between total content of phenols, chlorogenic and sinapic acids. Moreover, caffeic acid and DPPH scavenging are, as cited, a positive relationship. Finally, also negative correlations should be evidenced, such as total flavonoid content, sinapic and chlorogenic acids.
Lines 220-230: here, also the found correlations among the antimicrobial activities and the extract composition must be better explained and enforced by existing literature: from Figure 2, I see a negative relationship between FCR and Aspergillus sp. and between ferulic acid and S. aureus; on the other hand, levels of sinapic acid have a positive interaction with S. enteridis.
Lines 234-240: what was the sample dimension? Was it sampled in replicate? How many replicates?
Lines 243-247: this section needs some detail, especially about the management of the resinous, water-insoluble fraction. Were the samples treated by liposoluble solvent prior water extraction? Or, alternatively, was the resin eliminated by separation and centrifugation (or decantation) of the extract? Were the extract centrifuged (or only decanted), since in line 245 I see the word “supernatant”? In this case, details of the centrifugation process should be added. Please, clarify.
Line 260: this is very important. Did the extracts were hydrolized or not? In what form they were injected into HPLC system? This has to be clarified, it is not clear. The reason is that in plants, free phenols are difficult to be detected, because their “in planta” presence is under conjugated forms, mainly glicosidic ones. The Authors should explain how the detected phenols were identified with the given standards, were no conjugated phenols are evident, with the exception of chlorogenic acid, vitexin, and rutin. The alternative situation is that the phenols are in bounded forms with macromolecules, such as lignin (this is expecially for phenolic acids). In these both cases, the most used methodological approach is the characterization of the obtained extract after an acidic or alcaline hydrolisis, often performed with thermal treatments. Otherwise, the presence of free phenols in an acqueous extract is very low, unless the used samples, that I do not know for personal experience, are particular in this sense. Within this context, I suspect that, with very high levels of free caffeic and ferulic acid, for example, these compounds should come from an hydrolized extract. The Authors are invited to reply to this important issue and, if an hydrolitic approach has been used, its methodology should be added to the text.
Line 263: correct “C18”.
Line 273: it is very strange that the retention time of apigenin is close to the void volume of the column (1.10 minutes), and that of vitexin, its glicosidic derivative, is much more higher (8.00 minutes). In this reversed phase system, it should be the opposite situation. Please, clarify.
Line 288: in this case, a water-ethanol extract is reported to be used. Is it exact? I have understood that all analyses have been made on water extracts of coniferous shoots.
Line 290: correct “FeSO4·7H2O”.
Line 297: a clarification is needed regarding the DPPH calculation. If the final data has been given in inhibition percent, and the calibration was made by Trolox, was the inhibition plotted against Trolox data, to give the final results as μmoles Trolox equivalents per g of dried material, as indicated in Table 3? Please, clarify and correct, if possible.
Lines 315-316: please, clarify the used extracts for the microbiological assays. The acqueous extracts of coniferous shoots were used, and not the “liquid extract of ground hop cone medium”. Has it been pasted from another work?
Author Response
Dear Reviewer,
Thank you for all your comments and suggestions. We revised the manuscript, and we hope the manuscript meets the requirements for publication in the journal.
All responses to your comments are listed below.
Sincerely, Marcin Dziedzinski
Comments: Line 53: sorry, my mistake in the suggested correction: “… belongs to water-insoluble fraction.”
Line 70: correct “conifers”.
Line 106, Table 2: correct “kaempferol”.
Line 197: please, add the measure units of the given values 60-6390.
Line 201: please correct to “μmoles/g”, or “mM”, optionally.
Response: We corrected all these mistakes.
Comments: Lines 217-219: also if the analyzed solutions are complex in their phenolic composition, a deeper explanation enforced by comparison with existing literature of the calculated correlations with antioxidant indexes must be given. Very interesting are the found positive correlations between caffeic acid and gallic acid and between total content of phenols, chlorogenic and sinapic acids. Moreover, caffeic acid and DPPH scavenging are, as cited, a positive relationship. Finally, also negative correlations should be evidenced, such as total flavonoid content, sinapic and chlorogenic acids.
Lines 220-230: here, also the found correlations among the antimicrobial activities and the extract composition must be better explained and enforced by existing literature: from Figure 2, I see a negative relationship between FCR and Aspergillus sp. and between ferulic acid and S. aureus; on the other hand, levels of sinapic acid have a positive interaction with S. enteridis.
Response: We extended the discussion, and discussed correlation results.
Comment: Lines 234-240: what was the sample dimension? Was it sampled in replicate? How many replicates?
Response: I am not sure if that anwers your question, but we sampled shoots from three different trees, and took 10 g of needles from three different shoots of each species. From 30 g of ground needles we took sample of 5 g to prepare tested extracts. Then we tested three samples from each extract.
Comment: Lines 243-247: this section needs some detail, especially about the management of the resinous, water-insoluble fraction. Were the samples treated by liposoluble solvent prior water extraction? Or, alternatively, was the resin eliminated by separation and centrifugation (or decantation) of the extract? Were the extract centrifuged (or only decanted), since in line 245 I see the word “supernatant”? In this case, details of the centrifugation process should be added. Please, clarify.
Response: We conducted only water extraction as written in the manuscript, without using liposoluble solvent. Samples were only decanted, the word „supernatant” was improperly used in this case. It has been corrected.
Comment: Line 260: this is very important. Did the extracts were hydrolized or not? In what form they were injected into HPLC system? This has to be clarified, it is not clear. The reason is that in plants, free phenols are difficult to be detected, because their “in planta” presence is under conjugated forms, mainly glicosidic ones. The Authors should explain how the detected phenols were identified with the given standards, were no conjugated phenols are evident, with the exception of chlorogenic acid, vitexin, and rutin. The alternative situation is that the phenols are in bounded forms with macromolecules, such as lignin (this is expecially for phenolic acids). In these both cases, the most used methodological approach is the characterization of the obtained extract after an acidic or alcaline hydrolisis, often performed with thermal treatments. Otherwise, the presence of free phenols in an acqueous extract is very low, unless the used samples, that I do not know for personal experience, are particular in this sense. Within this context, I suspect that, with very high levels of free caffeic and ferulic acid, for example, these compounds should come from an hydrolized extract. The Authors are invited to reply to this important issue and, if an hydrolitic approach has been used, its methodology should be added to the text.
Response: The section on both acidic and basic hydrolysis has been supplemented.
Comment: Line 263: correct “C18”.
Response: It has been corrected
Comment: Line 273: it is very strange that the retention time of apigenin is close to the void volume of the column (1.10 minutes), and that of vitexin, its glicosidic derivative, is much more higher (8.00 minutes). In this reversed phase system, it should be the opposite situation. Please, clarify.
Response: Corrected retention times, a mistake caused editorial inaccuracies
Comment: Line 288: in this case, a water-ethanol extract is reported to be used. Is it exact? I have understood that all analyses have been made on water extracts of coniferous shoots.
Response: In this case, we tested water exctracts, we corrected that mistake.
Comment: Line 290: correct “FeSO4·7H2O”.
It has been corrected
Comments : Line 297: a clarification is needed regarding the DPPH calculation. If the final data has been given in inhibition percent, and the calibration was made by Trolox, was the inhibition plotted against Trolox data, to give the final results as μmoles Trolox equivalents per g of dried material, as indicated in Table 3? Please, clarify and correct, if possible.
There was a mistake in methodology, the results were measured against Trolox, and are expressed as in the Table 3.
Lines 315-316: please, clarify the used extracts for the microbiological assays. The acqueous extracts of coniferous shoots were used, and not the “liquid extract of ground hop cone medium”. Has it been pasted from another work?
Response: We use simmilar methodology across few different studies that we are conducting currently, sorry for that mistake. It has been corrected.